# The Impact of Cadmium Selenide Zinc Sulfide Quantum Dots on the Proteomic Profile of *Saccharomyces cerevisiae*

**DOI:** 10.3390/ijms242216332

**Published:** 2023-11-15

**Authors:** Nhi Le, Abhishu Chand, Onyinye Okafor, Kyoungtae Kim

**Affiliations:** Department of Biology, Missouri State University, Springfield, MO 65897, USA; nhi0407@live.missouristate.edu (N.L.); ac43s@missouristate.edu (A.C.); odo84s@missouristate.edu (O.O.)

**Keywords:** quantum dots, CdSe/ZnS QDs, proteomic, protein expression, toxicity

## Abstract

Quantum dots (QDs) have been highly sought after in the past few decades for their potential to be used in many biomedical applications. However, QDs’ cytotoxicity is still a major concern that limits the incorporation of QDs into cutting-edge technologies. Thus, it is important to study and understand the mechanism by which QDs exert their toxicity. Although many studies have explored the cytotoxicity of quantum dots through the transcriptomic level and reactive species generation, the impact of quantum dots on the expression of cellular protein remains unclear. Using *Saccharomyces cerevisiae* as a model organism, we studied the effect of cadmium selenide zinc sulfide quantum dots (CdSe/ZnS QDs) on the proteomic profile of budding yeast cells. We found a total of 280 differentially expressed proteins after 6 h of CdSe/ZnS QDs treatment. Among these, 187 proteins were upregulated, and 93 proteins were downregulated. The majority of upregulated proteins were found to be associated with transcription/RNA processing, intracellular trafficking, and ribosome biogenesis. On the other hand, many of the downregulated proteins are associated with cellular metabolic pathways and mitochondrial components. Through this study, the cytotoxicity of CdSe/ZnS QDs on the proteomic level was revealed, providing a more well-rounded knowledge of QDs’ toxicity.

## 1. Introduction

From commercial products to biomedical treatments, the use of nanoparticles in modern technologies has become more prevalent in the past few decades. Among the many types of nanoparticles, a fluorescence semiconductor nanocrystal called quantum dots (QDs) stood out for its potential to be used in a wide range of applications. QDs’ many distinguishable characteristics such as high quantum yield [1,2], narrow emission spectrum [3,4,5], and photobleaching-resistant fluorescence [6] allowed them to become a highly sought-after material for several biomedical technologies, such as disease detection [7,8,9,10,11] and drug delivery [12,13,14,15], as well as commercialized products such as electronic devices [16,17], LED light [18,19], and solar cells [20,21,22,23]. However, as QDs gain more attention for their potential in these applications, concerns have been raised regarding the use of QDs in these applications due to QDs’ potential toxicity.

In recent years, many researchers have reported the negative impact of QDs on different cellular systems as well as the mechanism of their toxicity [24,25,26,27]. It was found that QD exposure led to an alteration in the expression of many genes that play a key role in essential cellular processes [28,29,30]. In addition, QDs have also been reported to cause an elevation in reactive oxygen species and induce apoptosis-mediated cell death in different mammalian cell lines [31]. Furthermore, recently published articles have also proposed that QDs can exert toxicity through direct interaction with proteins and cause post-translation protein modifications [32]. Understanding the mechanism of QD toxicity is crucial to the development of safer QDs. Thus, examination of the different ways in which QDs exert toxicity is valuable and urgently needed.

Although many mechanisms of QDs’ cytotoxicity have been revealed, the effect of QDs on the proteomic profile of cells remained poorly investigated. As a stable protein level is needed for proper cellular functions, the lack of research on this matter creates a large knowledge gap in the field that limits our comprehensive understanding of QD toxicity. Therefore, in this study, we examined the impact of the commonly used cadmium selenide zinc sulfide quantum dots (CdSe/ZnS QDs) on the proteomic profile of *Saccharomyces cerevisiae*. Using the state-of-the-art nano LC-MS/MS technology, we identified a number of proteins whose expression levels were altered by CdSe/ZnS QDs. Furthermore, in this paper, we have also provided an in-depth classification of the altered proteins by protein class and functions, as well as pinpointed the cellular processes and pathways that were significantly impacted. Lastly, by measuring the protein expression of *Saccharomyces cerevisiae* in vivo, we were also able to verify our proteomic data and confirm the influence of CdSe/ZnS QDs on the protein expression in budding yeast.

## 2. Results

### 2.1. Proteomic Alterations by CdSe/ZnS QDs

Using liquid chromatography–mass spectrometry (LC-MS) analysis, we were able to assess the changes to the proteomic profile of *Saccharomyces cerevisiae* exposed to CdSe/ZnS QDs. Our results showed a total of 280 proteins whose expression was altered when comparing the treated samples to the non-treated control samples. Among them, 187 proteins were found to be upregulated (FC > 1.2, *p* < 0.05), and 93 proteins were found to be downregulated (FC < 1.2, *p* < 0.05).

Among the upregulated proteins, transcription/RNA processing-associated proteins (34), intracellular trafficking-associated proteins (29), and ribosome components and processes-associated proteins (27) are most abundant (Figure 1A). Other categories of upregulated proteins include unidentified (20), metabolism (18), mitochondrial components and processes (17), glycolysis (11), upregulates when stressed (10), proteasome/protein modification (7), chaperon (4), actin (4), DNA replication/repair (4), chromatin (2), and others (9).

On the other hand, a large number of downregulated proteins are associated with metabolism (27) and mitochondrial components (10) (Figure 1B). Other categories of downregulated proteins include unknown (10), ribosome components and processes (8), proteosome/protein modification (7), transcription/RNA processing (6), glycolysis (6), trafficking (4), actin (4), chaperon (3), cell cycle (3), cell wall (2), DNA repair (1), chromatin (1), and others (5).

To further illustrate the identified proteins, we created each diagram of differentially expressed proteins associated with intracellular trafficking and mitochondrial components (Figure 2A,B). For the intracellular trafficking diagram, a number of upregulated proteins were indicated, including endocytosis-associated proteins (ART5, YPP1, PIL1, SCP1, CMD1), endosome to Golgi trafficking (VPS54, MON2, DOP1, TRS120), Golgi fusion (COG6), Golgi to ER (GET1), nucleocytoplasmic transport (SEH1, NUP188, POM152, RPS3, NUP192, NMD3, NUP170, KAP123), ER to Golgi (RER1, SEC7, AGE1, EMP24), exocytosis (EXO7), and vacuole transport and fusion (ATG27, VTC3, IVY1). The identified mitochondrial proteins are outer membrane (PTH2), inner membrane (COX5A), intermembrane space (CCP1, CMC2, MIX17, MIX14), inner membrane (MDL1, AAC3, OXA1, OAC1, MRS4, ODC1, PET191), 50S ribosome (MRPL22, MRPL35), and nucleoid (ALD4).

### 2.2. Classification of Proteins Using Gene Ontology

To further analyze the identified proteins, we used the Gene Ontology (GO) function on the Saccharomyces Genome Database (SGD) website to categorize the proteins into specific cellular components (Figure 3A,B), as well as cellular processes (Figure 3C,D). The upregulated proteins were divided into twenty groups based on their specific cellular components (Figure 3A). However, these can be further simplified into six major categories: ribosome components and complexes (ribosome, ribosome subunits, cytosolic ribosome, ribonucleoprotein complex, pre-ribosome, 90S pre-ribosome), intracellular membrane-bound organelle components, intracellular non-membrane-bound components, lumen (membrane-enclosed lumen, organelle lumen, and nucleus lumen), cytosolic components, and other cellular complexes (TOR complex, TORC2 complex, macrocellular complex). Identified downregulated proteins were sorted into nine groups based on their cellular components by the SGD (Figure 3B). These groups could be further simplified into five major categories: mitochondrial components, intracellular organelle components, cytoplasmic components, cell and intracellular structure components, and organelle envelop lumen.

Using the SGD, we categorized the DEPs into different cellular processes. The upregulated proteins were placed into twenty processes (Figure 3C), which could be further simplified into six major categories: ribosome biogenesis (ribosomal small subunit, ribonucleoprotein complex, ribosome), RNA processing (ncRNA processing, SSU-rRNA maturation, rRNA processing), transportation (nucleolytic cytoplasmic transport, nucleic acid transport, nucleobase-containing compound transport), localization and organization of cellular components, cellular components biogenesis, metabolic process (rRNA and ncRNA metabolic process). The downregulated proteins were placed into twenty cellular process groups by the SGD (Figure 3D). These processes can be simplified into three major categories: metabolic processes (organonitrogen compound metabolic process, ADP metabolic process, ribonucleoside diphosphate biosynthetic process, drug metabolic process, organophosphate metabolic process, nucleobase-containing small molecule metabolic process, small molecule biosynthetic process, phosphate-containing compound metabolic process, purine nucleoside diphosphate metabolic process), protein modification (protein folding, peptidyl-proline modification, protein peptidyl-prolyl isomerization), and other cellular processes.

### 2.3. Enriched Pathways

Identification of pathways significantly impacted by the DEP is important for insightful assessments of the toxicity of CdSe/ZnS QDs. Using the KEGG pathway classification function on the Kobas website, we sorted the up- and downregulated proteins into different cellular pathways to find the ones that are most affected by the treatment of CdSe/ZnS QDs. For upregulated proteins, the pathways that are found to be most affected are ribosome biogenesis, RNA transport, starch and sucrose metabolism, RNA polymerase, and phenylalanine metabolism (Figure 4A). Analysis of the downregulated proteins revealed the fifteen pathways that are significantly affected are related to metabolism. These pathways are glycolysis/gluconeogenesis, biosynthesis of secondary metabolites, carbon metabolism, biosynthesis of antibiotics, biosynthesis of amino acids, phosphatidylinositol signaling system, purine metabolism, glycine/serine/threonine metabolism, thiamine metabolism, arginine biosynthesis, amino sugar and nucleotide sugar metabolism, fructose and mannose metabolism, one carbon pool by folate, riboflavin metabolism, and vitamin B6 metabolism (Figure 4B).

For the upregulated proteins, RNA transport was one of the pathways that was majorly impacted according to our KEGG analysis (Figure 4A). Upon further assessment, many of these proteins are involved in the nuclear pore complex (NPC) and the nucleocytoplasmic transport system (Figure 5). The NPC is located in the membrane of the nuclear envelope, and it acts as a gateway for the bidirectional exchange of materials, such as mRNA and proteins, between the nucleus and the cytoplasm [33]. The NPC and the nucleocytoplasmic transport process are essential for many cellular processes and functions. Defects in the NPC and the nucleocytoplasmic transport process are linked to several human diseases including neurodegenerative diseases such as Huntington’s disease [34,35] and Alzheimer’s disease [36]. Specific components with identified DEPs are the spoke complex, which is the framework of the nuclear pore complex [37]; the cytoplasmic ring, a component serving as a platform that helps with the import of cytoplasmic materials into the nucleus [38]; and the exportin complex, which is essential for the export of nuclear materials to the cytoplasm.

Glycolysis is an essential metabolic pathway as it generates ATP, the energy-carrying molecule needed for cellular processes, and other important intermediates such as pyruvate [39]. According to our KEGG pathway analysis, five enzymes in the glycolysis and glucogenesis pathway were significantly downregulated (Figure 6). These enzymes are phosphoglucose isomerase (PGI1); fructose-biphosphate aldolase (FBA1); triose phosphate isomerase (TPI1); 3-phosphoglycerate kinase (PGK1); and enolase II (ENO2). It is worth noting that several enzymes that participated in the glycolysis process were also found to be upregulated. The upregulated enzymes are Hexokinase isoenzyme 1 (HXK1); Early Meiotic Induction 2 (EMI2); Mitochondrial aldehyde dehydrogenase 4 (ALD4); and Mitochondrial aldehyde dehydrogenase 3 (ALD3). The alteration in the expression of essential glycolytic enzymes suggests that the treatment of CdSe/ZnS QDs leads to a lower production of ATP, which may result in the deprivation of the energy needed for the operation of other cellular processes.

### 2.4. Verification of Proteomic Data

In our initial classification of DEPs in Section 2.1, twenty-nine identified proteins were intracellular trafficking-associated proteins. Among these proteins, Pil1, a protein involved in eisosome assembly and endocytosis, was found to be overexpressed (fold change: 1.33, *p*-value: 0.013). To verify our proteomic data, we used the budding yeast cells expressing Pil1 tagged with green fluorescence protein (Pil1-GFP) to measure the level of Pil1 protein expression upon treatment of CdSe/ZnS QDs (Figure 7A,B). The overexpression of the Pil1 protein is consistent with our proteomic data.

## 3. Discussion

In this paper, the alteration in the proteomic profile of *Saccharomyces cerevisiae* treated with CdSe/ZnS QDs was revealed. Upon 6 h of CdSe/ZnS QDs incubation, several DEPs were identified. Notably, many upregulated proteins are associated with transcription/RNA transport, intracellular trafficking, and ribosome biogenesis. On the other hand, downregulated proteins are mostly identified as essential metabolic proteins and proteins that are associated with the structure and function of the mitochondria.

In 2019, Horstmann et al. examined the transcriptomic profile of *Saccharomyces cerevisiae* in the presence of CdSe/ZnS QDs. Their results showed that although metabolic genes were both up- and downregulated in response to CdSe/ZnS QDs, most of the metabolic genes were found to be downregulated [29]. Furthermore, the same paper also revealed an upregulation of the genes in the RNA transport and ribosome biogenesis pathway, similar to our findings at the proteomic level. In contrast, while our data revealed an enrichment of the proteins in many of the intracellular trafficking processes, Horstmann et al. found a reduction in the expression of many endocytosis genes [29]. Recent studies have reported that the size of quantum dots plays a key role in QD toxicity, in which quantum dots with smaller sizes are more toxic [40,41]. Although both studies used water-soluble CdSe/ZnS QDs quantum dots with carboxylic ligands obtained from NN-Lab, Horstmann et al. used yellow-emitting CdSe/ZnS QDs (CZW-Y) with a total diameter of 7.2 nm while our red-emitting CdSe/ZnS QDs (CZW-R) have a total diameter of around 9 nm. Thus, the contradictory results in the expression of the endocytic genes found by Horstmann et al. and our proteomic study might reside in the difference in the size of the used quantum dots and their mechanism of toxicity.

Recently, Gallo et al. have investigated the proteomic alteration of *Saccharomyces cerevisiae* induced by cadmium sulfide quantum dots (CdS QDs). Similar to our results, Gallo et al. found a number of DEPs in the glycolysis and glucogenesis pathway [42]. Interestingly, Fba1 was the only common downregulated glycolytic protein identified by both Gallo et al. and our study. This suggests that downregulation of the glycolysis pathway may be a common toxicity mechanism for different types of quantum dots. However, the specific protein differentially expressed is dependent on the quantum dots type. An altered glycolysis process can have dire consequences due to the inefficient production of energy and intermediate metabolites needed for other key cellular processes. In addition, Ghosh et al. reported the ability of quantum dots to interfere with the enzymatic activity of glyceraldehyde-3-phosphate dehydrogenase, a glycolysis enzyme, through site-specific interaction in vitro [43]. Thus, it is possible that quantum dots with varying composition and surface ligands could bind to different glycolysis enzymes, causing further inefficiency of glycolysis, and resulting in more deprivation of the energy.

The toxicity of quantum dots is complex and varies with the composition, size, and other properties. Thus, the need to investigate the cytotoxicity of different quantum dots on a proteomic level is still needed. Additionally, the proteomic profile of other cellular systems in response to quantum dots should also be examined for a more comprehensive understanding of quantum dots’ cytotoxicity.

## 4. Materials and Methods

### 4.1. Characterization of CdSe/ZnS QDs

Cadmium selenide zinc sulfide quantum dots (CZW-R-5) with carboxylic ligand, suspended in water (1 mg/mL), with a total diameter of around 9 nm were obtained from NN-Labs (Fayetteville, AR, USA). The emission peak of these quantum dots is around 620–650 nm according to the manufacturer’s specifications. Zhang et al. (our lab) have previously characterized the air-dried diameter of these quantum dots to be 5–10 nm, while the hydrodynamic diameter is approximately 20 nm [24], which is consistent with the manufacturer’s information.

### 4.2. Yeast Strain Preparation

The wildtype (S288c) strain of *Saccharomyces cerevisiae* was streaked and grown on yeast extract peptone dextrose or YPD agar plate (2% dextrose, peptone, yeast extract, 2% agar). After 2 days, one isolated colony was transferred into a fresh YPD medium and incubated at 30 °C overnight. On the next day, the optical density of the fresh yeast stock was measured and diluted until reaching the density of 0.1. Then, the diluted yeast culture was divided into 6 flasks, each containing 25 mL of the diluted yeast culture. Three of the flasks were used as non-treated control and three of the flasks were treated with 25 µg/mL of CdSe/ZnS QDs. The samples were then incubated for 6 h at 30 °C. Next, the samples were spun at 3000 rpm for 10 min to obtain the cell pellets. Cell pellets were then shipped on dry ice to Creative Proteomic (Shirley, NY, USA) for LC-MS/MS proteomic analysis.

### 4.3. Protein Extraction and Identification by Liquid Chromatography–Mass Spectrometry

Samples were suspended in BPP buffer (100 mM EDTA, 50 mM borax, 50 mM Vitamin C, 30% Sucrose, 100 mM Tris base, 1% TritonX-100, 1% to 5% PVPP, 5 mM DTT, pH 8.0) supplied with 1× protease inhibitor cocktail and phosphate inhibitor. The samples were then placed in the low-temperature tissue grinding machine. Samples were ground four times at −40 °C (shock 120 s, stop 60 s). An equal volume of Tris-saturated phenol (pH 8.0) was added and vortexed for 10 min and then centrifuged for 12,000× *g* for 20 min at 4 °C (Tris-saturated phenol was handled according to chemical safety requirements). The supernatant was then collected. Five volumes of cold 0.1 M ammonium acetate in methanol were added to precipitate protein overnight at −20 °C. The samples were centrifuged, and the pellet was collected and then washed twice with 90% acetone. The pellet was then resuspended with lysis buffer (1% sodium deoxycholate, 8 M urea) with a protease inhibitor cocktail. The lysate was centrifuged, and the supernatant was collected. The concentration of protein was tested by the BCA (bicinchoninic acid) Protein Assay Kit. An amount of 100 µg of protein was then reduced with 2 µL of 0.5 M Tris(2-carboxyethyl)phosphine (TCEP) for 60 min at 30 °C and then alkylated with 4 µL 1 M iodoacetamide (IAM) at room temperature for 40 min away from light. Protein was precipitated overnight. The protein pellet was resuspended with 100 µL 10 mM triethylammonium bicarbonate (TEAB) buffer. Trypsin (Promega, Madison, WI) was added at 1:50 trypsin-to-protein mass ratio and incubated at 37 °C overnight. Protein samples were then lyophilized.

Before use, the peptide samples were labeled with TMT (tandem mass tag) labeling agents. The peptide mixture was then re-dissolved in buffer A (buffer A: 20 mM ammonium formate in water, pH 10.0, adjusted with ammonium hydroxide), and then fractionated by high pH separation using Ultimate 3000 system (ThermoFisher Scientific, Waltham, MA, USA) connected to a reverse-phase column XBridge C18 column, 4.6 mm × 250 mm, 5 μm (Waters Corporation, Milford, MA, USA). High pH separation was performed using a linear gradient, starting from 5% B to 45% B in 40 min (B: 20 mM ammonium formate in 80% ACN, pH 10.0, adjusted with ammonium hydroxide). The column was re-equilibrated at the initial condition for 15 min. The column flow rate was maintained at 1 mL/min and the column temperature was maintained at 30 °C. Ten fractions were collected; each fraction was dried in a vacuum concentrator for the next step.

The mass spectrometer was run under data-dependent acquisition (DDA) mode, and automatically switched between MS and MS/MS (tandem mass spectrometry) mode. The survey of full-scan MS spectra (*m*/*z* 350–1500) was acquired in the Orbitrap with 120,000 resolution. The normalized automatic gain control (AGC) target was 200% and the maximum injection time was 50. Then, the precursor ions were selected into the collision cell for fragmentation by higher-energy collision dissociation (HCD); the normalized collection energy was 33. The MS/MS resolution was set at 30,000, with the normalized automatic gain control (AGC) target of 200%, the maximum injection time of 54ms, and dynamic exclusion was 30 s.

The 10 raw MS files were analyzed and searched against the yeast protein database based on the species of the samples using Maxquant (2.1.0.0). The parameters were set as follows: the protein modifications were carbamidomethylating (C) (fixed), oxidation (M) (variable), TMT-16plex; the enzyme specificity was set to trypsin; the maximum missed cleavages were set to 2; the precursor ion mass tolerance was set to 10 ppm; and MS/MS tolerance was 0.5 Da.

### 4.4. Initial Characterization of DEPs

Identified proteins were sorted into upregulated protein (fold change > 1.20, *p*-value < 0.05), downregulated protein (fold change < 0.81, *p*-value < 0.05), or no change. The up- and downregulated proteins were identified and categorized into protein classes using the SGD website as a reference. The protein class was then graphed with Prism GraphPad 10.

### 4.5. Gene Annotation of the DEPs Using SGD

The up- and downregulated proteins were subjected to the GO term finder function on the SGD website and sorted by components and processes. The components and processes identified were then graphed using Prism GraphPad 10.

### 4.6. KEGG Pathway Classification

The up- and downregulated proteins were categorized into pathways using the enrichment function on the Kobas website (Kobas.cbi.pku.edu.cn, accessed on 28 June 2023). The gene names of the identified up- or downregulated proteins were imported and the KEGG pathway function was chosen. After obtaining the results, only pathways with *p*-values less than 0.05 were selected. Data of the pathways, enrichment ratio, and *p*-values were inserted into the pathway enrichment bubble plot function on the bioinformatic website (Bioinformatics.com.cn/en, accessed on 30 June 2023).

### 4.7. Measurement of Pil1-GFP Expression Using Line Intensity Assay

Yeast strains KKY0201 were streaked on selective agar of Synthetic Defined media lacking Histidine or SD-His (2% glucose, yeast nitrogen base with amino acids, complete supplement mixture minus histidine, 2% agar). The plate was then incubated stationary at 30 °C. For every experiment, fresh cultures were prepared in the following manner: one individual colony was transferred to 3 mL of SD-His liquid media and grown in a shaker incubator for 24 h at 30 °C to create a fresh stock. The optical density (OD) at 600 nm was measured and the cell concentration was diluted to achieve an optical density of 0.1. Subsequently, the cells were treated and cultured with CdSe/ZnS QDs at concentrations of either 25 µg/mL or 50 µg/mL for 6 h at a temperature of 30 °C. Cells were then visualized using the spinning confocal microscope, and two-dimensional still images were recorded. Thirty (n = 30) eisosomes in each treatment were analyzed to measure the fluorescence intensity of Pil1-GFP and the width of the eisosome. Briefly, a peak line intensity was analyzed to determine the fluorescence intensity of Pil1-GFP using a Slidebook (v6), and then ImageJ was used to determine the width of the eisosome carrying Pil1-GFP. The numerical values obtained from the Slidebook (v6) were exported to Prism Graphpad 10 Using the column graph, the ANOVA non-parametric tab was employed to analyze the difference between the non-treated and two groups of treated samples. To determine the width of the eisosomes, still images were exported to ImageJ and analyzed using the “plot profile” icon. We randomly selected 30 puncta to determine the average width of the eisosome in each treatment. The eisosome width was calculated by multiplying the pixel number by 0.133 µm (133 nm/pixel). Statistical analysis was conducted by using Prism Graphpad 10. For analysis of the eisosome marker (See Section 2.4), the program stated above was launched and the column graph was selected. The imported values obtained from the Slidebook and ImageJ from the analysis of the fluorescence intensity and width of the eisosome marker, respectively, were appropriately labeled. A “two-tailed”, “non-parametric”, and “Dunnett’s multiple comparison” icons were selected, and a one-way ANOVA test was employed to analyze the variance between the non-treated and two groups of treated samples. On each prism graph generated, each sample is represented by an average of thirty patches and has error bars representing standard deviation. Statistically significant data are represented on graphs as * *p* < 0.05, ** *p* < 0.01, *** *p* < 0.001, and **** *p* < 0.0001.

## Figures and Tables

**Figure 1 ijms-24-16332-f001:**
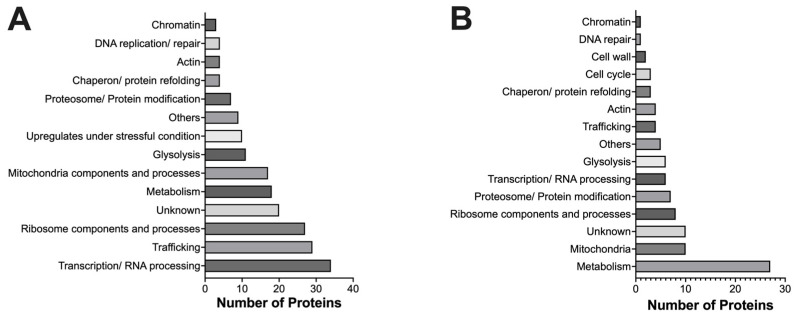
Classification of protein demonstrating altered expressions upon the treatment of CdSe/ZnS QDs. (**A**) Classification of proteins that were upregulated by the treatment of CdSe/ZnS QDs. (**B**) Classification of proteins that were downregulated by the treatment of CdSe/ZnS QDs.

**Figure 2 ijms-24-16332-f002:**
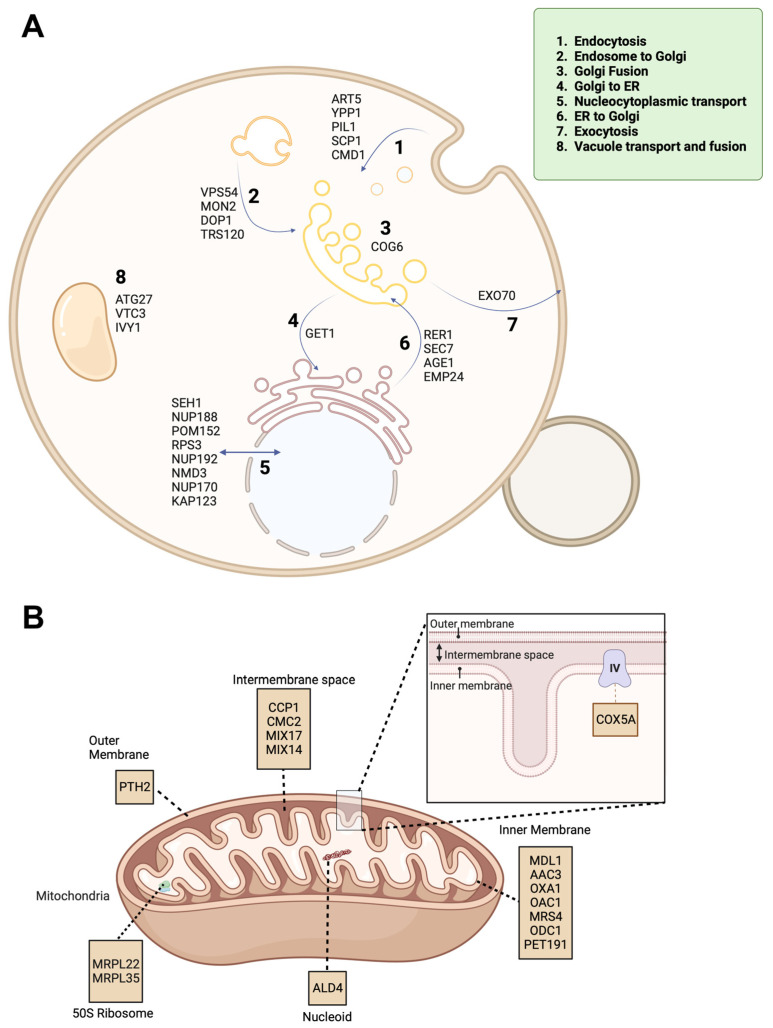
Diagrams of differentially expressed proteins (DEPs). (**A**) DEPs associated with intracellular trafficking pathways. (**B**) DEPS associated with mitochondrial components. Created with BioRender.com.

**Figure 3 ijms-24-16332-f003:**
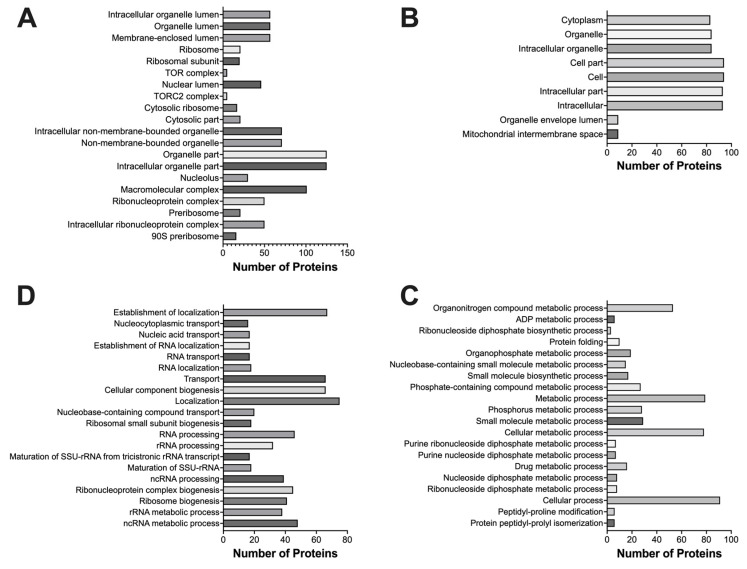
Classification of DEPs based on cellular components and processes using SGD. (**A**) Classification of upregulated proteins based on cellular components. (**B**) Classification of downregulated proteins based on cellular components. (**C**) Classification of upregulated proteins based on cellular processes. (**D**) Classification of downregulated proteins based on downregulated cellular processes.

**Figure 4 ijms-24-16332-f004:**
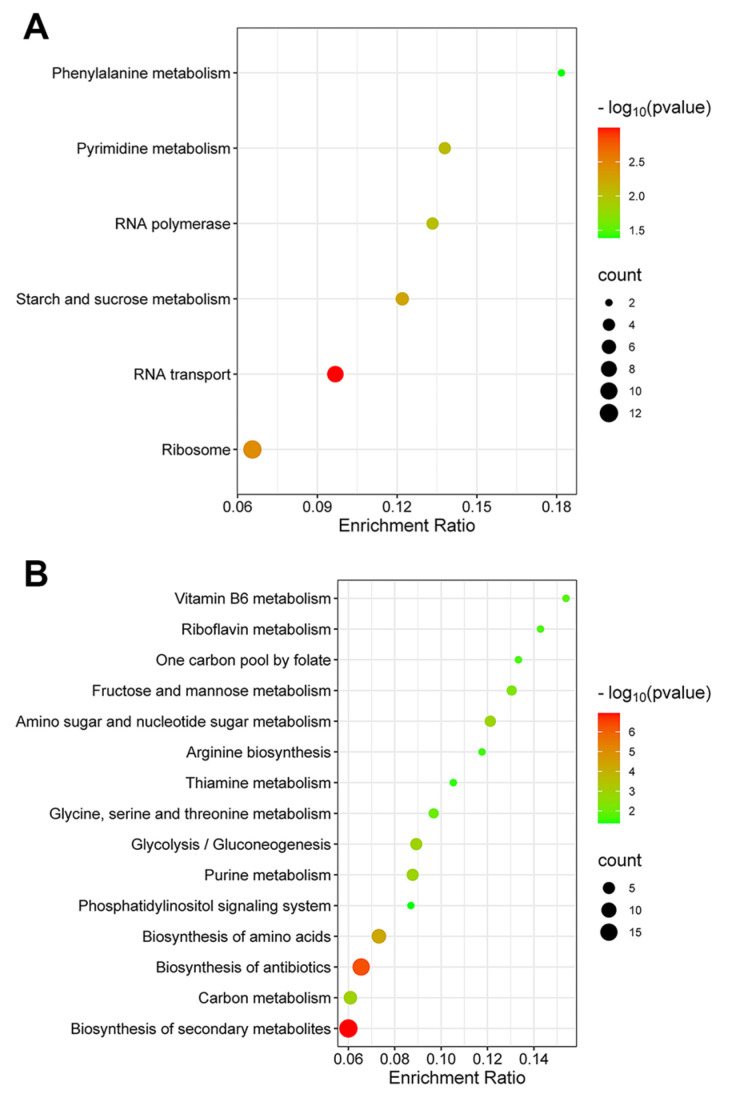
Pathways are significantly enriched with DEPs. The size of each dot represents the number of DEPs in each pathway. The color of each dot represents the *p*-values in terms of −log_10_. The enrichment ratio is equal to the number of DEPs divided by the number of expected proteins in each category. (**A**) Pathways significantly enriched with upregulated proteins. (**B**) Pathways significantly enriched with downregulated proteins.

**Figure 5 ijms-24-16332-f005:**
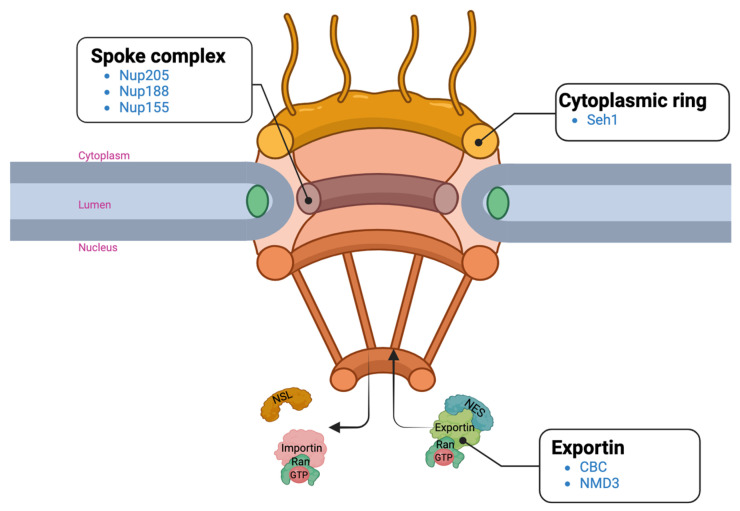
Diagram of the nucleocytoplasmic transport system with DEPs. Components with altered proteins are labeled in bold, black font and the gene names of the upregulated proteins are in blue font. Created with BioRender.com.

**Figure 6 ijms-24-16332-f006:**
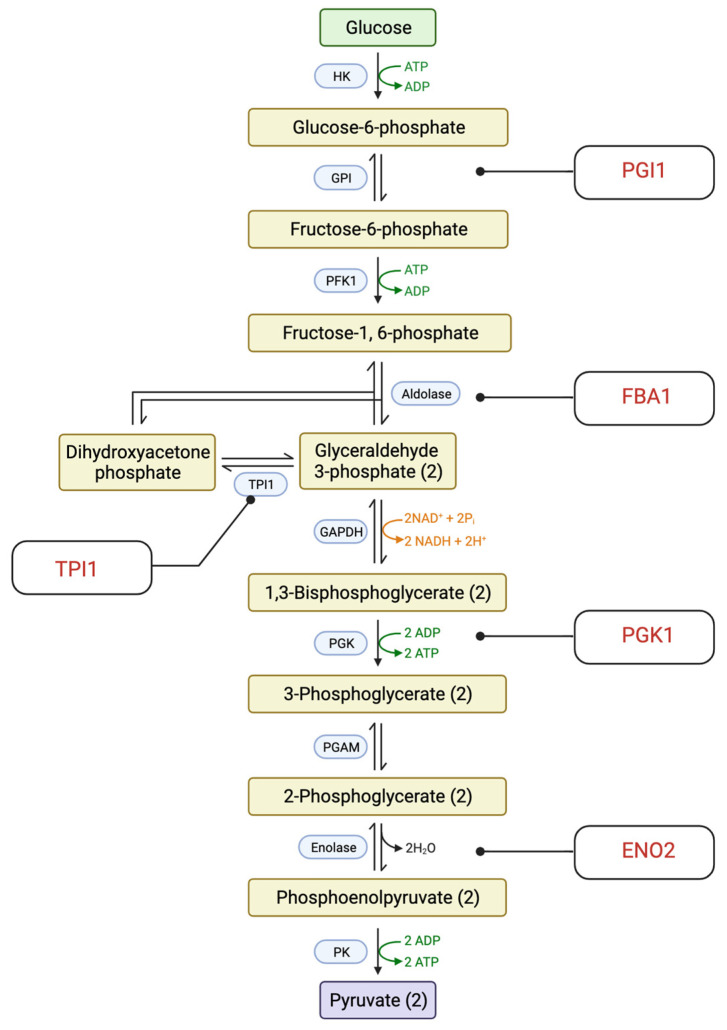
A diagram of the glycolysis and glucogenesis pathway. The gene name of the downregulated proteins is in red font. Created with BioRender.com.

**Figure 7 ijms-24-16332-f007:**
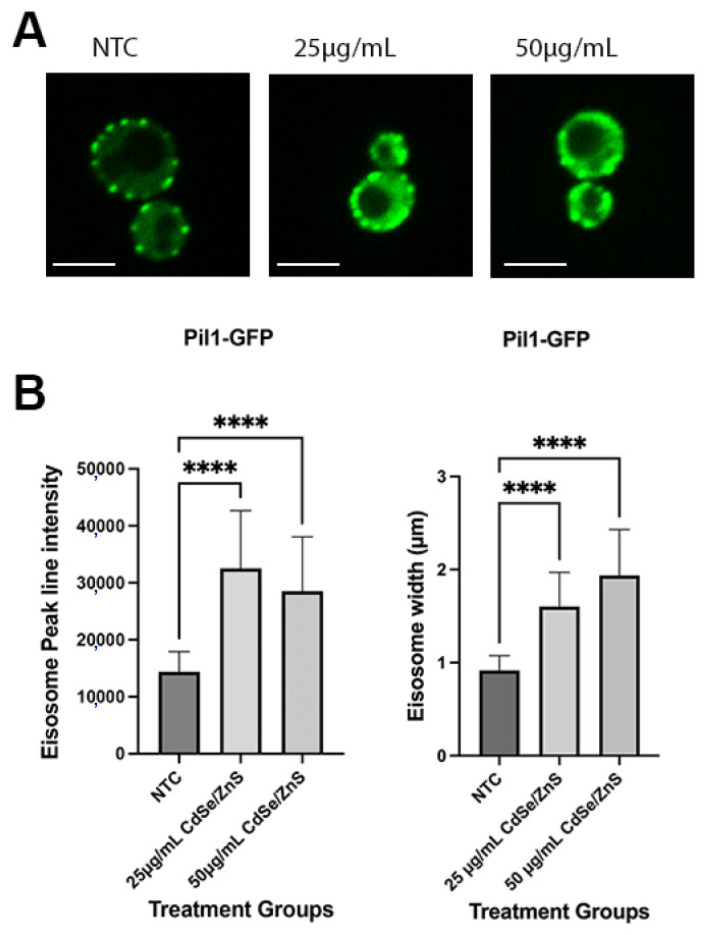
The expression of Pil1-GFP with and without the presence of CdSe/ZnS QDs. (**A**) Representative image of cells from samples with no quantum dots treatment (NTC), treated with 25 µg/mL of CdSe/ZnS QDs, or treated with 50 µg/mL of CdSe/ZnS QDs. The size bar is equivalent to 5 µm. (**B**) The measurement of the expression of Pil1-GFP by the intensity and eisosome width (µm). **** *p* < 0.0001.

## Data Availability

Data are contained within the article.

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
