# Peer review of "The Impact of Cadmium Selenide Zinc Sulfide Quantum Dots on the Proteomic Profile of Saccharomyces cerevisiae"

_ijms, 2023, doi:10.3390/ijms242216332_

Round 1
Reviewer 1 Report
Comments and Suggestions for Authors
This is a very interesting paper, reporting CdSe/ZnS QD toxicity. The topic is of the outmost importance, since cadmium is a well-known carcinogen and is also endowed with neurotoxicity. Moreover, QD size allows their quick diffusion within cells making them a big threat for human health. I recommend accepting this work, with a minor revision.
My main concern is that in paragraph 2.1, authors state that proteins associated to glycolysis are found both down and up regulated; however upregulated glycolytic proteins are not mentioned in the following paragraphs. The same is true for downregulated proteins associated with mitochondrial components and processes, mentioned in paragraph 2.1. Which glycolytic enzymes are upregulated and which mitochondrial proteins/enzymes are downregulated? Authors should discuss the relationship between these altered processes; is it possible that key glycolytic enzymes are upregulated following oxidative phosphorylation downregulation? Or that both processes are impaired?
Minor points are:
Line 88 – “upregulated proteins were into twenty…” a verb is missing; maybe “upregulated proteins were divided into…”
Line 107 – substitute “simply” with “simplified”
Line 114 – again substitute “simply” with “simplified”
Line 164 – enolase, not enolse
Author Response
Please see the attached letter

Reviewer 2 Report
Comments and Suggestions for Authors
In this work of “The Impact of Cadmium Selenide Zinc Sulfide Quantum Dots on the Proteomic Profile of Saccharomyces cerevisiae” authors identified and classified the number of proteins altered by CdSe/ZnS QDs and demonstrated its influence on protein expression in yeast culture.
Quantum dots are commercially largely used products. But to mitigate all un-accepted toxicity related hazards, more deep knowledge is necessary to be scientifically accumulated and used in practice. Because of that reason, this study is relevant to be added to the growing body of knowledge of Quantum dots. And manuscript would be interesting to IJMS readers.
In general, the manuscript is well written, the information presented is well organized. The methods could be written more clearly and avoided repetitions.
I’d like the authors to address the main suggestions/comments:
55: classification of the altered proteins by their protein class and functions – this needs to re-phrased. “By their proteins” is confusing.
62: “LC-MS” – define it Liquid chromatography/mass spectrometry.
63: “5 µg/mL of CdSe/ZnS QDs for 6 hrs.” this information is provided in the methods and repeated there three times.
96: “SGD” – define it (Saccharomyces Genome Database)
176: ”Pil1-GFP” – define it - green fluorescent protein (GFP)
237: “The wildtype (S288c) strain of Saccharomyces cerevisiae was streaked and grown on yeast extract peptone dextrose (YPmass spectrometry.D) agar plate “– write percentage of agar and refer the product.
238: “After 2 days, an isolated colony was inoculated into a fresh YPD medium” – I guess, you mean one individual colony was picked and transferred to fresh liquid medium for incubation? And write incubation temperature.
239-240: “25 mL of the fresh yeast culture was used to create three non-treated control samples and three CdSe/ZnS QDs treated samples (concentration 25 ug/mL)” – this needs to be re-phrased.
⁻ You should write what was volume in ml for each control and test samples. What does mean 25 ml? did you use 4.17 ml for each of six? And why is this volume important?
⁻ Create is not correct world in this case. I guess you made equal aliquots from fresh east liquid culture.
⁻ Write treatment conditions: time, temperature…
245: “4.3. Protein extraction and Identification” – refer the method
250-251: “an equal volume of Tris-saturated phenol (pH 250 8.0) was added and vortexed for 10 min and then centrifuged for 12000 g for 20 mins at 4” –
Here you should indicate that “Tris-saturated phenol” was handled under chemical fume hood and out of chemical hood safety requirements were followed.
257: “BCA Protein Assay Kit” –write BCA stands for “bicinchoninic acid”
264: “TMT “ - here as well (tandem mass tag)
276 “MS/MS” – here as well (Tandem mass spectrometry)
303: “After obtaining the results, only pathways with p-values less than 0.05 were obtained”- needs to be re-phrased. It is confusing obtained results obtained”
307: “4.7. Measurement of Pil1-GFP expression” – refer the method.
308-309: “Yeast strains KKY0201 were streaked on appropriate selective agar plate Synthetic Defined nutrient lacking Histidine (SD-His) and incubated for a few days in a 30 ℃ stationary incubator”.
⁻ “Appropriate”- doesn’t say anything here – write “selective agar of synthetic defined media lacking histidine”; capital letters are not appropriated here. And refer the product.
⁻ Was incubated stationary at 30 ℃.
310: “Before each experiment, an isolated colony from each plate was selected, moved to 3 mL of corresponding liquid media” – this is confusing. I guess, for every experiment fresh culture were prepared: one individual colony was transferred to 3 ml liquid media. Refer the media.
312-314: The optical density (OD) at 600 nm was recorded, and cells were diluted to make an optical density of 0.1 at 0 hours and cultured for 6 hours in the presence or absence of CdSe/ZnS-COOH QDs. Cells were treated and incubated either with 25 µg/mL or 50 µg/mL of CdSe/ZnS QDs for 6 hours at 30 ℃ - repetitions – rephrase it.
It is repeated above as well pages 239-240.
Comments on the Quality of English LanguageModerate editing of English language required
Author Response
Please see the attached letter.
